# Study of Skin Barrier Function in Psoriasis: The Impact of Emollients

**DOI:** 10.3390/life11070651

**Published:** 2021-07-04

**Authors:** Daniel Maroto-Morales, Trinidad Montero-Vilchez, Salvador Arias-Santiago

**Affiliations:** 1Dermatology Department, Faculty of Medicine, University of Granada, 18071 Granada, Spain; danielmarotomorales@gmail.com (D.M.-M.); salvadorarias@ugr.es (S.A.-S.); 2Department of Dermatology at Hospital Universitario Virgen de las Nieves, 18012 Granada, Spain; 3Instituto de Investigación Biosanitaria ibs.GRANADA, 18012 Granada, Spain

**Keywords:** emollients, homeostasis, moisturizers, psoriasis, skin physiology, skin barrier

## Abstract

Psoriasis is a chronic multi-systemic inflammatory disease that affects the epidermal barrier. Emollients can be used as a coadjutant therapy for psoriasis management, but little is known about how the epidermal barrier function in psoriatic patients is modified by moisturizers. The objective of this study is to evaluate the effect of Vaseline jelly and a water-based formula on epidermal barrier function in psoriatic patients. Thirty-one patients with plaque-type psoriasis and thirty-one gender and age-matched healthy controls were enrolled in the study. Temperature, transepidermal water loss (TEWL), stratum corneum hydration (SCH), pH, elasticity and the erythema index were measured using non-invasive tools before and after applying Vaseline jelly and a water-based formula. TEWL was higher in psoriatic plaques than uninvolved psoriatic skin (13.23 vs. 8.54 g·m^−2^·h^−1^; *p* < 0.001). SCH was lower in psoriatic plaques than uninvolved psoriatic skin and healthy skin (13.44 vs. 30.55 vs. 30.90 arbitrary units (AU), *p* < 0.001). In psoriatic plaques, TEWL decreased by 5.59 g·m^−2^·h^−1^ (*p* = 0.001) after applying Vaseline Jelly, while it increased by 3.60 g·m^−2^·h^−1^ (*p* = 0.006) after applying the water-based formula. SCH increased by 9.44 AU after applying the water-based formula (*p* = 0.003). The use of emollients may improve epidermal barrier function in psoriatic patients. TEWL is decreased by using Vaseline, and SCH is increased by using the water-based formula.

## 1. Introduction

Psoriasis is a chronic, immune-mediated inflammatory disease that affects 1–3% of the world’s population [1]. Prevalence rates show worldwide geographic variation from 0.51% (USA) to 11.43% (Norway) [2]. The most common manifestations of psoriasis are scattered, erythematous, scaly papules and plaques. Nevertheless, this condition presents different skin patterns, and may even affect several joints and cause changes in the nails. Its etiology remains unclear, even though numerous risk factors have been related to its predisposition or pathology, such as epigenetic alterations, obesity, alcohol consumption, smoking and stress [3,4]. Moreover, psoriasis causes great psychosocial impact [5] and is related to many organic comorbidities, such as cardiovascular disease [6].

The skin is the largest organ of the human body and carries out multiple defensive and regulatory functions. The loss of skin integrity due to injury or illness may result in a substantial physiological imbalance, disability and in some cases death [7]. The epidermal barrier is important for protecting the human body against many external stressors, and for maintaining skin homeostasis. Transepidermal water loss (TEWL) is regarded as one of the most important parameters for measuring the integrity of the skin barrier. It is defined as the flux density of water that diffuses from the epidermis and dermis through the stratum corneum on the skin’s surface. Increased TEWL levels are associated with skin barrier impairments [8,9,10]. Stratum corneum hydration (SCH) is another important parameter for assessing skin barrier function. It shows the water content of the stratum corneum, and lower SCH values are frequently associated with dermatological conditions and greater disease severity [11]. Other skin characteristics related to skin barrier function include pH, elasticity, temperature and erythema index [12].

Moisturizers play an important role in skin barrier repair. Emollients improve the barrier function of the stratum corneum, providing water and lipids. This lipid replacement therapy may reduce inflammation and restore epidermal function [13]. Moisturizers are mainly composed of saturated and unsaturated hydrocarbons of variable length. They also contain occlusive ingredients, such as petrolatum or lanolin, that coat the skin surface with a water-repellent lipid layer, impeding the bidirectional movement of water (mostly by reducing skin water loss) [14].

Regarding psoriasis, the role of emollients is important as an adjuvant therapy, but their impact on the epidermal barrier is not well known. They help to normalize hyperproliferation, differentiation and apoptosis, in addition to having anti-inflammatory effects. Emollients reduce scaling and itching, soften cracks and improve the penetration of other topical drugs [15,16]. The objective of this study is to evaluate the impact of emollients on epidermal barrier function in patients with psoriasis.

## 2. Materials and Methods

### 2.1. Design

A cross-sectional study to assess skin homeostasis differences between healthy skin, involved and uninvolved psoriatic skin.

A before and after study on patients with psoriasis, in order to assess changes in skin barrier function after applying two different emollients.

### 2.2. Study Sample

Participants were recruited from July to October 2020 at the Dermatology Service of the Hospital Universitario Virgen de las Nieves in Granada, Spain.

#### 2.2.1. Inclusion Criteria:

Patients with psoriasis and an established clinical diagnosis of mild to severe plaque-type psoriasis [17].Healthy volunteers were gender and age matched (±5 years) with the cases, who attended the Dermatology Service for common conditions, such as seborrheic keratoses or melanocytic nevi, and did not have a previous history of psoriasis.Aged between 18 and 65 years old.

#### 2.2.2. Exclusion Criteria:

Psoriasis patients that have a type of psoriasis different from plaque-type psoriasis, such as guttate psoriasis, pustular psoriasis or inverse psoriasis.Failing to sign the informed consent form.

### 2.3. Emollients

Two emollients were tested: pure Vaseline jelly and a water-based formula. Pure Vaseline jelly was 100% Vaseline composition (mixture of semisolid hydrocarbons), without any excipients, in an ointment pharmacologic form. The water-based formula was composed of the emulsifier base NEO PCL O/W (with low-fat content) 18 g, distilled water 39 g, Phenonip 0.18 g, glycerol 3 g.

### 2.4. Variables

Homeostasis parameters related to epidermal barrier function were measured. TEWL (in g·m^−2^·h^−1^, using Tewameter^®^ TM 300, Courage + Khazaka electronic GmbH, Köln, Germany), SCH (in arbitrary units, using Corneometer^®^ CM 825, Courage + Khazaka electronic GmbH, Köln, Germany), skin temperature (in °C, using Skin-Thermometer ST 500, Courage + Khazaka electronic GmbH, Köln, Germany), pH (measured in pH units, using Skin-pH-Meter PH 905, Courage + Khazaka electronic GmbH, Köln, Germany), skin elasticity (using R2 value measured in %, using Cutometer^®^ Dual MPA 580, Courage + Khazaka electronic GmbH, Köln, Germany) and erythema index (in arbitrary units, using Mexameter^®^ MX 18, Courage + Khazaka electronic GmbH, Köln, Germany) were measured by Multi Probe Adapter (MPA, Courage + Khazaka electronic GmbH, Köln, Germany). Elasticity parameters were measured four times, and the other variables were measured ten times, using their average for analysis. Psoriatic patients were measured at a psoriatic plaque site on the elbow and an uninvolved psoriatic skin area (on a similar anatomic location on the opposite side of the body), while controls were only measured on the elbow. Measurements were taken at each location twice: the first measurement before applying the emollients, and the second one 20 min after applying both emollients (Vaseline jelly and water-based formula). The distance between skin spots treated with the two different emollients was 5 cm, a sufficient distance to ensure there is no interaction with the treatments tested. The same amount of each moisturizer (0.05 mg) was applied to each area (psoriatic plaques, uninvolved psoriatic skin and healthy skin) and measurements were taken again after 20 min. The measurements were taken in the same room. All participants underwent an adaptation period of at least 20 min before the first measurements were taken. The average ambient air temperature and ambient humidity at the time of the study were 24 ± 3 °C and 42 ± 2%, respectively.

Clinical variables were also evaluated. Psoriasis severity was assessed by the psoriasis area and severity index (PASI) and the body surface area (BSA). All patients were also evaluated with the dermatology life quality index (DLQI). Gender, age, age at diagnosis, family history of psoriasis, psoriatic arthropathy, smoking habit, alcohol intake, current treatments and regular use of emollients were gathered by means of clinical interview. Some anthropometric measures were also collected, such as weight (kg), height (m), body mass index (BMI) and abdominal perimeter (cm).

### 2.5. Statistical Analysis

Categorical variables were expressed as relative and absolute frequencies, and continuous variables as the mean (standard deviation). The Shapiro–Wilk test was used to check the normality of data distribution. Linear regression models were constructed to compare continuous data between healthy skin and psoriatic patients. Categorical data were compared using Pearson’s chi-squared test. Continuous independent variables were contrasted using Student’s *t*-test for independent variables. To compare homeostasis parameters before and after applying the emollient, a Student’s *t*-test for paired samples was used. A *p*-value of <0.05 was considered statistically significant. Statistical analyses were performed using the SPSS package (SPSS for Windows, Version 24.0 Chicago: SPSS Inc., Chicago, IL, USA).

## 3. Results

### 3.1. Sample Characteristics

A total of 62 subjects were included in the study, consisting of 31 psoriatic patients and 31 healthy controls. Table 1 summarizes the general characteristics of the sample. 54.8% (34/62) were male and 45.2% (28/61) were female. The mean age was 50.50 (15.82) years old, without any differences between healthy volunteers and patients with psoriasis.

Psoriatic patients were more likely to be smokers than healthy volunteers (41.9% vs. 12.9%, *p* = 0.010). The BMI was higher in the psoriasis group (30.88 vs. 24.01 kg/m^2^, *p* < 0.001), and the abdominal perimeter was also higher in psoriatic patients (110.58 vs. 90.42 cm, *p* < 0.001).

### 3.2. Skin Homeostasis Analysis between Psoriatic Plaques, Uninvolved Psoriatic Skin and Healthy Controls

Skin barrier function parameters were compared between healthy volunteers, involved and uninvolved skin in psoriatic patients before any emollients were applied (Table 2).

TEWL was significantly higher in psoriatic plaques than uninvolved psoriatic skin (13.23 vs. 8.54 g·m^−2^·h^−1^, *p* < 0.001) and healthy skin (6.41 g·m^−2^·h^−1^, *p* = 0.021). SCH was significantly lower in psoriatic plaques than both uninvolved psoriatic skin (13.44 vs. 30.55 AU, *p* < 0.001) and healthy controls (30.90 AU, *p* < 0.001). Temperature was higher in psoriatic plaques than at healthy skin (30.68 vs. 29.71 °C, *p* = 0.01). pH was significantly lower in psoriatic plaques than healthy controls (6.26 vs. 6.60, *p* = 0.02). The erythema index was higher in psoriatic plaques than on healthy skin (380.40 vs. 307.63 AU, *p* < 0.001). No differences in elasticity were found.

### 3.3. Skin Homeostasis Changes after Applying Emollients

Homeostasis parameters changed after applying emollients, as shown in Figure 1.

On uninvolved psoriatic skin (Table 3), TEWL increased by 2.75 g·m^−2^·h^−1^ after applying the water-based formula (12.21 vs. 8.542 g·m^−2^·h^−1^; *p* = 0.02), while TEWL decreased by 1.43 g·m^−2^·h^−1^ after applying Vaseline jelly (8.03 vs. 8.54 g·m^−2^·h^−1^, *p* = 0.056). Differences were also found between both emollients on TEWL parameters (8.03 vs. 12.21 g·m^−2^·h^−1^; *p* < 0.001). SCH increased after the water-based formula was applied (38.08 vs. 30.55 AU; *p* = 0.05). No differences in SCH values were found after Vaseline jelly application. A higher SCH increase was observed after water-based formula application than after Vaseline jelly use (38.08 vs. 28.58 AU; *p* < 0.001). Temperature decreased by 0.33 °C (*p* = 0.03) after applying the water-based formula, but it did not change after Vaseline jelly application. No differences in pH, erythema or elasticity were found after applying moisturizers.

In psoriatic plaques (Table 4), TEWL decreased by 5.59 g·m^−2^·h^−1^ (*p* = 0.001) after applying Vaseline jelly while it increased by 3.60 g·m^−2^·h^−1^ (*p* = 0.006) after applying the water-based formula. SCH increased by 9.44 AU after applying the water-based formula (*p* = 0.003) while it did not change after applying Vaseline. Temperature decreased by 0.44 °C (*p* = 0.017) after applying the water-based formula, whereas it did not change after Vaseline. Erythema index increased by 59.33 AU (*p* < 0.001) after Vaseline application and by 58.16 AU (*p* < 0.001) after the water-based formula application. No differences in pH or elasticity were found after applying emollients.

On healthy skin (Table 5), TEWL decreased by 0.99 g·m^−2^·h^−1^ (*p* = 0.025) after Vaseline jelly application and increased by 2.83 g·m^−2^·h^−1^ (*p* = 0.010) after water-based formula application. SCH increased by 7.33 AU (*p* < 0.001) after applying the water-based formula while it did not change after applying Vaseline. Temperature decreased by 0.49 °C (*p* < 0.001) after applying Vaseline jelly and by 0.59 °C (*p* < 0.001) after the water-based formula. Erythema index decreased by 16.62 AU (*p* = 0.012) after applying the water-based formula but did not change after applying Vaseline. No differences in pH or elasticity were found after applying emollients.

## 4. Discussion

TEWL, temperature and erythema were higher in psoriatic plaques than uninvolved psoriatic skin and health controls, while SCH and pH were lower. Vaseline jelly application reduced TEWL in psoriatic plaques, uninvolved psoriatic skin and healthy skin. Applying the water-based formula increased SCH and decreased temperature both in psoriatic plaques, uninvolved psoriatic skin and healthy skin.

Differences in skin barrier function parameters between healthy skin, uninvolved skin and psoriatic plaques were observed in psoriasis patients. Psoriatic plaques showed higher TEWL and erythema values, as well as lower SCH values than uninvolved skin and healthy skin. TEWL was also significantly higher in uninvolved psoriatic skin than healthy skin. This reflects that the whole epidermal barrier is affected in psoriatic patients, not just at psoriatic plaques. Other research also showed higher TEWL in psoriatic plaques than uninvolved psoriatic skin and healthy skin [10,12,18,19]. Most studies also found that psoriatic plaques have higher TEWL values than uninvolved psoriatic skin [10,12,18]. Regarding SCH values, other studies showed lower SCH levels in psoriatic plaques than non-involved skin [10,12,19,20], supporting our results. Authors have explained that the differences in TEWL and SCH values between psoriatic plaques and non-involved skin in the same subject is due to a decrease in AQP3 expression in plaques and perilesional skin [21]. These high TEWL values and lower SCH values are translating a skin barrier dysfunction in psoriatic patients, both on psoriatic plaques and uninvolved psoriatic skin [8,9,10].

There is less information regarding other skin barrier function parameters. Our study showed that psoriatic patients had higher skin temperature and lower pH values than healthy controls, both in plaques and uninvolved skin. Previously, controversial results have been reported for pH values in psoriatic patients. Similar to our results, Cannavò et al. found lower pH values for psoriatic plaque and uninvolved skin [20]. However, Delfino et al. reported no difference in this parameter [22]. Our research also found higher temperature and erythema values in psoriatic skin, which could be explained by its inflammatory pathogenesis [23]. Changes in elasticity in psoriasis have only been evaluated by Choi et al. who showed lower values for psoriatic patients assessed by R7 parameter [24], in contrast to similarities in elasticity values observed in our report. These differences could be explained by the difference in elasticity measures between studies. Choi et al. assessed elasticity using R7, ratio of elastic recovery to total deformation, while we used R2, overall elasticity. It has been reported that R2 is a more reliable elasticity parameter than R7 [25]. Our previous research did not find any differences in elasticity either [10,12].

Sex, age, BMI and smoking habit are factors that could impact on skin barrier function. There are skin physiological distinctions between sex and age [26,27]. Moreover, it has been found that obese children have higher TEWL values than normal weight children [28] and that smoking habit damages skin barrier function [29]. We chose healthy individuals that were gender and age matched (±5 years) with psoriatic patients to avoid the influence that sex and age might have on the differences. As we observed differences at baseline in BMI and smoking habit between psoriatic patients and healthy individuals, a linear regression model adjusted by BMI and smoking habit was constructed, to control for the differences introduced by these factors. In that way, the differences observed in skin barrier function between psoriatic patients and healthy individuals in this study are not influence by age, sex, BMI or smoking habit.

There is little research on the role of emollients in epidermal barrier function in psoriasis. Our results show improved epidermal barrier function at the psoriatic plaque reflected in TEWL reduction after Vaseline jelly application, and an increase in SCH after applying the water-based formula. Changes in TEWL after using moisturizers are controversial in the literature. Draelos et al. did not find any changes in TEWL values after 4-weeks’ treatment with a moisturizer in 30 psoriatic patients [30]. As far as we know, there are no more studies regarding the impact of emollients on epidermal barrier function in psoriatic patients. In atopic dermatitis (AD), TEWL improved in 20 AD patients after 4 weeks of treatment with four different moisturizers [31]. Nevertheless, Mohammed et al. found increased TEWL values after applying emollients for 28 days [32]. This inconsistent data on the effect of moisturizers on TEWL is possibly explained by the different composition of the moisturizers used in each study [31]. In agreement with this data, we observed a decrease in TEWL values after Vaseline jelly application, which may be explained by its immediate barrier-repairing effect in delipidized stratum corneum [14]. Nevertheless, the water-based formula increased TEWL, which may be explained by its water-only composition, which is not occlusive enough to change this parameter [33]. Both emollients increased SCH in psoriatic plaques. In agreement with our results, Draelos et al. evaluated 30 patients with psoriasis who received a moisturizing cream for 4 weeks and also observed an increase in skin hydration [30]. Higher increases in SCH using water-based formulas may be due to the presence of humectants, such as glycerine, that are not contained in Vaseline or petrolatum.

The reduction in temperature after the water-based formula application on psoriatic skin may be explained by the increased TEWL values. When TEWL increases, the skin temperature decreases slightly [34]. Differences in temperature values between both moisturizers could also be related to the nature of the moisturizer. Erythema rising at psoriatic plaques after application is explained by a physical effect. Applying the emollient removes the dry scales that covered the lesion surface, displaying a deep, more erythematous layer of the plaque [14].

This study was subject to several limitations. (1) The variability of skin homeostasis parameters depends on external conditions; nevertheless, to increase outcome reliability, all participants were measured in the same room and the ambient conditions were also measured. (2) Concomitant systemic and topic medication was allowed, giving the possibility of strong variability between subjects; however, paired samples were used to assess the effect of applying emollients, meaning that confounding factors regarding intra-individual characteristics were controlled (each participant is compared with themselves). In this way, concomitant treatments would not change the results. (3) Only two moisturizers with different compositions were tested. Further research should be conducted regarding the impact of other emollients on skin barrier function in psoriasis patients, to compare them and assess more beneficial components for repairing skin barrier function, not only in psoriasis but also in other dry skin conditions.

## 5. Conclusions

In conclusion, emollients improved the epidermal barrier function in psoriatic patients. SCH increased after applying Vaseline jelly and the water-based formula. TEWL decreased after applying Vaseline jelly. The analysis of the cutaneous homeostasis parameters might help us understand the role of emollients in improving psoriatic patients’ care, recommend specific moisturizers to improve skin barrier function and thus improve psoriatic symptoms and quality of life.

## Figures and Tables

**Figure 1 life-11-00651-f001:**
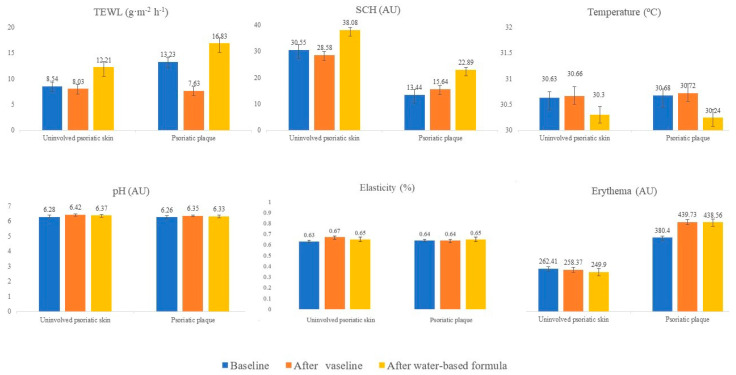
Changes in homeostasis parameters after using emollients.

**Table 1 life-11-00651-t001:** Characteristics of the sample.

	All Participants (*n* = 62)	Controls (*n* = 31)	Psoriatic Patients (*n* = 31)	*p* *
Age (years)	50.50 (±15.82)	47.77 (±16.19)	53.23 (±15.21)	0.559
Sex (%)				1
- Female	28 (45.2%)	14 (45.2%)	14 (45.2%)	
- Male	34 (54.8%)	17 (54.8%)	17 (54.8%)	
Smoking habit				0.010
- Non-smoker	45 (72.6%)	27 (87.1%)	18 (58.1%)	
- Smoker	17 (27.4%)	4 (12.9%)	13 (41.9%)	
Alcohol intake (excessive)	4 (6.5%)	1 (3.2%)	3 (9.7%)	0.301
Family history of psoriasis (yes)	17 (27.4%)	3 (9.7%)	14 (45.2%)	0.002
Emollient use (yes)	27 (43.5%)	11 (35.5%)	16 (51.6%)	0.2
Weight (kg)	79.13 (±18.06)	69.10 (±11.73)	89.15 (±17.84)	<0.001
Height (m)	1.70 (±0.11)	1.69 (±0.11)	1.70 (±0.11)	0.773
BMI (kg/m^2^)	27.48 (±5.95)	24.013 (±2.83)	30.88 (±6.29)	<0.001
Abdominal perimeter (cm)	100.50 (±17.21)	90.42 (±12.04)	110.58 (±15.72)	<0.001
DLQI			5.81 (±4.82)	
PASI			5.23 (±3.78)	
BSA			6.41 (±4.91)	
Psoriatic arthritis (yes)			15 (48.4%)	
Current treatment				
- Topical			19 (61.3%)	
- Oral medication			6 (19.4%)	
- Phototherapy			5 (16.1%)	
- Biologic drugs			14 (45.5%)	

BMI, body mass index; DLQI, dermatology life quality index; PASI, psoriasis area severity index; BSA, body surface area. * *p* value after using Student’s *t*-test for independent samples or Welch’s test when required to compare continuous variables; and the chi-square test or Fisher’s exact test, as appropriate, to compare categorical data between control and psoriatic patients.

**Table 2 life-11-00651-t002:** Homeostasis parameters in controls versus uninvolved psoriatic skin and psoriatic plaques, before applying emollients.

	Control	Uninvolved Psoriatic Skin before Applying Emollients	Psoriatic Plaque before Applying Emollients	*p* *	*p* **	*p* ***
TEWL (g·m^−2^·h^−1^)	6.41 (±4.41)	8.54 (±3.87)	13.23 (±7.85)	0.022	0.021	<0.001
SCH (AU)	30.90 (±12.22)	30.55 (±11.78)	13.44 (±14.17)	0.073	<0.001	<0.001
Temperature (°C)	29.71 (±1.19)	30.63 (±1.75)	30.68 (±2.11)	0.007	0.010	0.812
pH	6.60 (±0.36)	6.28 (±0.51)	6.26 (±0.51)	0.002	0.002	0.674
Elasticity (%)	0.6423 (±0.16)	0.6288 (±0.17)	0.6361 (±0.16)	0.904	0.838	0.853
Erythema (AU)	307.63 (±55.05)	262.41 (±55.23)	380.40 (±96.41)	0.031	0.002	<0.001

AU, arbitrary units; SCH, stratum corneum hydration; TEWL, transepidermal water loss * *p* value after using a linear regression model adjusted by smoking habit and BMI to compare homeostasis parameters between control and uninvolved psoriatic skin before applying emollients. ** *p* value after using a linear regression model adjusted by smoking habit and BMI to compare homeostasis parameters between control and psoriatic plaques before applying emollients. *** *p* value after using Student’s *t*-test for paired samples to compare homeostasis parameters between uninvolved psoriatic skin and psoriatic plaques before applying emollients.

**Table 3 life-11-00651-t003:** Homeostasis parameters at uninvolved psoriatic skin before and after applying different emollients.

	Uninvolved Psoriatic Skin before Applying Emollients	Uninvolved Psoriatic Skin after Applying Vaseline Jelly	Uninvolved Psoriatic Skin after Applying Water-Based Formula	Mean Difference at Uninvolved Skin before and after Applying Vaseline Jelly	Mean Difference at Uninvolved Skin before and after Applying Water-Based Formula	*p* *	*p* **	*p* ***
TEWL (g·m^−2^·h^−1^)	8.54 (±3.87)	8.03 (±3.60)	12.21 (±5.11)	−1.43 ( 4.01)	2.75 (±4.58)	0.056	0.020	<0.001
SCH (AU)	30.55 (±11.78)	28.58 (±11.71)	38.08 (±10.73)	−1.96 (±11.39)	7.54 (±13.76)	0.345	0.050	<0.001
Temperature (°C)	30.63 (±1.75)	30.66 (±1.94)	30.30 (±1.90)	0.03 (±0.79)	−0.33 (±0.79)	0.823	0.030	0.012
pH	6.28 (±0.51)	6.42 (±0.40)	6.37 (±0.39)	0.13 (±0.50)	0.09 (±0.45)	0.138	0.290	0.358
Elasticity (%)	0.6288 (±0.17)	0.6656 (±0.17)	0.6508 (±0.16)	0.0367 (±0.17)	0.0219 (±0.15)	0.240	0.413	0.589
Erythema (AU)	262.41 (±55.23)	258.37 (±51.95)	249.90 (±61.10)	−4.04 (±25.92)	−12.51 (±34.32)	0.392	0.051	0.147

AU, arbitrary units; SCH, stratum corneum hydration; TEWL, transepidermal water loss * *p* value after using Student’s *t*-test for paired samples to compare homeostasis parameters between uninvolved psoriatic skin before and after applying Vaseline jelly. ** *p* value after using Student’s *t*-test for paired samples to compare homeostasis parameters between uninvolved psoriatic skin before and after applying the water-based formula. *** *p* value after using Student’s *t*-test for paired samples to compare mean differences in homeostasis parameters between uninvolved psoriatic skin after applying both emollients.

**Table 4 life-11-00651-t004:** Homeostasis parameters in psoriatic plaques before and after applying different emollients.

	Psoriatic Plaque before Applying Emollients	Psoriatic Plaque after Applying Vaseline Jelly	Psoriatic Plaque after Applying Water-Based Formula	Mean Difference at Uninvolved Skin before and after Applying Vaseline Jelly	Mean Difference at Uninvolved Skin before and after Applying Water-Based Formula	*p* *	*p* **	*p* ***
TEWL (g·m^−2^·h^−1^)	13.23 (±7.85)	7.63 (±5.21)	16.83 (±6.86)	−5.59 (±5.68)	3.60 (±6.86)	<0.001	0.006	<0.001
SCH (AU)	13.44 (±14.17)	15.64 (±10.59)	22.89 (±14.66)	2.20 (±10.75)	9.44 (±16.23)	0.264	0.003	<0.001
Temperature (°C)	30.68 (±2.11)	30.72 (±1.87)	30.24 (±1.92)	0.04 (±0.90)	−0.44 (±0.96)	0.813	0.017	<0.001
pH	6.26 (±0.51)	6.35 (±0.35)	6.33 (±0.45)	0.09 (±0.40)	0.07 (±0.33)	0.229	0.277	0.685
Elasticity (%)	0.64 (±0.16)	0.64 (±0.18)	0.65 (±0.13)	0.00 (±0.21)	0.010 (±0.18)	0.977	0.762	0.664
Erythema (AU)	380.40 (±96.41)	439.73 (±90.14)	438.56 (±84.40)	59.33 (±53.83)	58.16 (±66.88)	<0.001	<0.001	0.912

AU, arbitrary units; SCH, stratum corneum hydration; TEWL, transepidermal water loss * *p* value after using Student’s *t*-test for paired samples to compare homeostasis parameters between psoriatic plaques before and after applying Vaseline jelly. ** *p* value after using Student’s *t*-test for paired samples to compare homeostasis parameters between psoriatic plaques before and after applying the water-based formula. *** *p* value after using Student’s *t*-test for paired samples to compare mean differences in homeostasis parameters between psoriatic plaques after applying both emollients.

**Table 5 life-11-00651-t005:** Homeostasis parameters in control skin before and after applying different emollients.

	Control Skin before Applying Emollients	Control Skin after Applying Vaseline Jelly	Control Skin after Applying Water-Based Formula	Mean Difference at Uninvolved Skin before and after Applying Vaseline Jelly	Mean Difference at Uninvolved Skin before and after Applying Water-Based Formula	*p* *	*p* **	*p* ***
TEWL (g·m^−2^·h^−1^)	6.41 (±4.41)	5.42 (±4.36)	9.23 (±4.98)	−0.99 (±2.32)	2.83 (±4.07)	0.025	0.010	<0.001
SCH (AU)	30.90 (±12.22)	30.48 (±11.68)	38.23 (±12.42)	−0.42 (±10.02)	7.33 (±7.21)	0.817	<0.001	<0.001
Temperature (°C)	29.71 (±1.19)	29.22 (±1.22)	29.12 (±1.23)	−0.49 (±0.81)	−0.59 (±1.06)	0.002	0.004	0.523
pH	6.60 (±0.36)	6.73 (±0.25)	6.57 (±0.29)	0.13 (±0.42)	−0.02 (±0.37)	0.095	0.720	0.013
Elasticity (%)	0.64 (±0.16)	0.65 (±0,13)	0.60 (±0.1433)	0.01 (±0.16)	−0.48 (±0.14)	0.851	0.066	0.012
Erythema (AU)	307.63 (±55.05)	297.23 (±58.69)	291.01 (±53.15)	−10.40 (±31.14)	−16.62 (±34.80)	0.073	0.012	0.197

AU, arbitrary units; SCH, stratum corneum hydration; TEWL, transepidermal water loss. * *p* value after using Student’s *t*-test for paired samples to compare homeostasis parameters between healthy skin before and after applying Vaseline jelly. ** *p* value after using Student’s *t*-test for paired samples to compare homeostasis parameters between healthy skin before and after applying water-based formula. *** *p* value after using Student’s *t*-test for paired samples to compare mean differences in homeostasis parameters between healthy skin after applying both emollients.

## Data Availability

The data presented in this study are available on request from the corresponding author.

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
