# Peer review of "Study of Skin Barrier Function in Psoriasis: The Impact of Emollients"

_life, 2021, doi:10.3390/life11070651_

Round 1
Reviewer 1 Report
The authors studied the effect of two emollients on the skin barrier in patients with psoriasis, on both involved and non-involved skin in comparison to subjects with healthy skin.
General comments:
Manuscript needs English proofreading due to number of typos. Some sentences are not clear (for example see line 64-65). Materials and method (measurements) should be written in a more ordered and more clear way.
Material and methods
2.2.2. Exclusion Criteria:
As one of the exlusion criteria was that psoriasis patients had non-plaque forms of psoriasis at the time of the study, but in section 2.4. you stated that emolients were applied to psoriatic plaques). How do you explain that?
Line 94-95: It might be applropriate to state what are the main ingrediences of “NEO PCL O/W” as you did for Vaseline jelly.
2.4. Variables
Line 104-106: it is not clear what the authors refered to when saying „located on the symmetrically opposite side of the body“. It is also not clear what was distance between skin spots treated with two different emollients? If they were close, next to each other, they might influence each other.
Additionally, why did you measure the effect of emollients at one time point and only after 20 minutes? How do you explain that this is the right time to evaluate the effect of emollients? It would be interesting to see for how long emollient are capable of maintaining these changes from one-time application.
Line 107-109: also not clear what the meaning, „Measures were taken at each location twice: the first one before emollients application, and the second one between 20 minutes after both emollients’ applications (vaseline jelly and water-based formula). This reads as you have applied both emmolients on the same spot, which I doubt.
Line 110: 0.05 mg (separate number from units)
Were subjects acclimatized to temperature/humidity in the room before any the measurements?
Results
In Table 2 you have shown p-value after using a linear regression model adjusted by smoking habit and BMI. What is the relevance of these results? After showing these results in the Table 2, authors did not address this point at all in the discussion! Authors should elaborate on this in the discussion section. Why did you adjust only BMI/smoking habits?
In table 3, “***p value after using Student’s T test for paired samples to compare homeostasis parameters between uninvolved psoriatic skin after both emollients application”. Which data did you exactly tested? Mean differences of absolute values obtained after application of emollients? This comments is applicable to table 4 and 5.
Disccusion
Discusion section is more of a repeating the results and their comparison to several other studies. What I mis is a discussion of results in terms of why, what would be reason for these effects. How do you explain their effect of skin barrier. What is the role of BMI and smoking habits (see comments above)
Line 309-312: first you say that TEWL after application of water-based formula did not changed, and in next sentence you say that TEWL increased. Which one is it?
I am not sure if paired analysis can fully compensate for concomitant systemic and even more topical application of medication during the study. They might change the course of the effect. Have you checked that to se if there is a difference between patients who use medication and those who do not us ed it at the time of the study?
Author Response
Reviewer 1
The authors studied the effect of two emollients on the skin barrier in patients with psoriasis, on both involved and non-involved skin in comparison to subjects with healthy skin.
General comments:
Manuscript needs English proofreading due to number of typos. Some sentences are not clear (for example see line 64-65). Materials and method (measurements) should be written in a more ordered and more clear way.
Thank you for the comments. Charlotte Bower, an English native speaker, specialized in scientific translation has reviewed carefully the manuscript. Material and methods had been reviewed and tried to be written in a more ordered and clear way.
Material and methods
2.2.2. Exclusion Criteria:
As one of the exlusion criteria was that psoriasis patients had non-plaque forms of psoriasis at the time of the study, but in section 2.4. you stated that emolients were applied to psoriatic plaques). How do you explain that?
We mean that we only included patients with plaque-type psoriasis. We excluded patients with other types of psoriasis (Guttate psoriasis, pustular psoriasis or Inverse psoriasis). Following your recommendations, we have changed this sentence in the exclusion criteria.
Line 94-95: It might be applropriate to state what are the main ingrediences of “NEO PCL O/W” as you did for Vaseline jelly.
The water-based formula was composed of the emulsifier base NEO PCL O/W (with low-fat content) 18g, distilled water 39g, phenonip 0.18g, glycerol 3g.
2.4. Variables
Line 104-106: it is not clear what the authors refered to when saying „located on the symmetrically opposite side of the body“. It is also not clear what was distance between skin spots treated with two different emollients? If they were close, next to each other, they might influence each other.
We wanted to say that it was measured on a similar anatomic location on the opposite side of the body. We have rephrased this sentence in the manuscript. We have added the distance between skin pots treated with the two emollients. The following sentence has been added: The distance between skin spots treated with the two different emollients (vaseline jelly and water-based formula) was 5cm, sufficient distance so that there is no interaction with the treatments tested.
Additionally, why did you measure the effect of emollients at one time point and only after 20 minutes? How do you explain that this is the right time to evaluate the effect of emollients? It would be interesting to see for how long emollient are capable of maintaining these changes from one-time application.
In this study we evaluated the short-term effect of the moisturizers which can be assessed between 15-30 after the application (Loden M, 2012; Lodén M et al 1992). We agree that it would be interesting to see for how long emollient are capable of maintaining these changes from one-time application but this study would contain many biases as the temperature, humidity, products employed for each participants could also influence in the emollients effect. Nevertheless, it is a really interesting idea and we would consider it for further research.
Line 107-109: also not clear what the meaning, „Measures were taken at each location twice: the first one before emollients application, and the second one between 20 minutes after both emollients’ applications (vaseline jelly and water-based formula). This reads as you have applied both emmolients on the same spot, which I doubt.
To clarify, we have changed the sentence: Measures were taken at each location twice (before and 20 minutes after emollients application).
Line 110: 0.05 mg (separate number from units)
We have separated the number from units.
Were subjects acclimatized to temperature/humidity in the room before any the measurements?
Yes, they were. We have added the following sentence in the manuscript: All participants underwent an adaptation period of at least 20 min before the first measurement was taken.
Results
In Table 2 you have shown p-value after using a linear regression model adjusted by smoking habit and BMI. What is the relevance of these results? After showing these results in the Table 2, authors did not address this point at all in the discussion! Authors should elaborate on this in the discussion section. Why did you adjust only BMI/smoking habits?
In table 3, “***p value after using Student’s T test for paired samples to compare homeostasis parameters between uninvolved psoriatic skin after both emollients application”. Which data did you exactly tested? Mean differences of absolute values obtained after application of emollients? This comments is applicable to table 4 and 5.
We compared the mean differences after emollients application. Following your recommendation, we have included this information in this both tables: *** p value Student’s T test for paired samples to compare mean differences in homeostasis parameters between psoriatic plaques after both emollients application. *** p value Student’s T test for paired samples to compare mean differences in homeostasis parameters between healthy skin after both emollients application.
Disccusion
Discusion section is more of a repeating the results and their comparison to several other studies. What I mis is a discussion of results in terms of why, what would be reason for these effects. How do you explain their effect of skin barrier. What is the role of BMI and smoking habits (see comments above)
We have included more information in the discussion to explain why changes are observed in our population and what is the role of BMI and smoking habits. The following sentences has been added: Some authors explained that the differences in TEWL and SCH values between psoriatic plaques and non-involved skin in the same subject is due to a decrease in AQP3 expression in plaques and perilesional skin[21]. This high TEWL values and lower SCH values are translating a skin barrier dysfunction in psoriatic patients both on psoriatic plaques and uninvolved psoriatic skin….Our research also found higher temperature and erythema values on psoriatic skin, which could be explained by its inflammatory pathogenesis… Sex, age, BMI and smoking habit are factors that could impact on skin barrier function. There are skin physiological distinctions between sexes and ages[26, 27]. Moreover, it has been found that obese children have higher TEWL values than normal-weight children[28] and that smoking habit damage skin barrier function[29]. We chose healthy individuals gender-and-age-matched (±5 years) with psoriatic patients to avoid that sex and age might influence on the differences. As we observed differences at baseline in BMI and smoking habit psoriatic patients and healthy individuals, a linear regression model adjusted by BMI and smoking habit was constructed to control the differences by these factors. In that way, the differences observed in skin barrier function between psoriatic patients and healthy individuals in this study are not influence by age, sex, BMI or smoking habit.”…
We observed a decrease in TEWL values after Vaseline jelly application what may be explained by its immediate barrier-repairing effect in delipidized stratum corneum[14]. This inconsistency data on the effect of moisturizers on TEWL is probably explained by the different composition of the moisturizers used in each study…. Nevertheless, water-based formula did not change TEWL, explained by its only water composition, not occlusive enough to revert this parameter[33].
Line 309-312: first you say that TEWL after application of water-based formula did not changed, and in next sentence you say that TEWL increased. Which one is it?
Sorry for the mistake. TEWL increased after water-based formula. We have rephase this sentence: Nevertheless, water-based formula increased TEWL, perhaps due to the water content of the master formula itself, not occlusive enough to change this parameter.
I am not sure if paired analysis can fully compensate for concomitant systemic and even more topical application of medication during the study. They might change the course of the effect. Have you checked that to se if there is a difference between patients who use medication and those who do not us ed it at the time of the study?
All our psoriatic patients were receiving treatment as is shown in table 1. We compared if there were differences depending on the type of treatment patients were receiving but we did not find any difference. Moreover, as we used a paired analysis, when comparing the impact of emollients, each participant is his own control, so the treatment should not influence on the differences observed.

Reviewer 2 Report
Thank you for your original contribution. This was an interesting and well-organized manuscript. Some remarks follow hereafter:
- Reference 3: J Am Acad Dermatol. 2021;84(1):46-52.
- Reference 10: J Clin Med. 2021;10(2):359.
- Reference 12: Please cite the stage of publication, i.e “in press” or the DOI:10.1111/phpp.12650.
- Although comprehensible, the manuscript would benefit from a careful grammar check.
Author Response
Thank you for your original contribution. This was an interesting and well-organized manuscript. Some remarks follow hereafter:
Thank you for the comments.
- Reference 3: J Am Acad Dermatol. 2021;84(1):46-52.
This reference has been checked.
- Reference 10: J Clin Med. 2021;10(2):359.
This reference has been checked.
- Reference 12: Please cite the stage of publication, i.e “in press” or the DOI:10.1111/phpp.12650.
Epub ahead of print has been added .
- Although comprehensible, the manuscript would benefit from a careful grammar check.
Charlotte Bower, an English native speaker, specialized in scientific translation has reviewed carefully the manuscript
Round 2
Reviewer 1 Report
No further comments.